# Modelling amoebic brain infection caused by *Balamuthia mandrillaris* using a human cerebral organoid

**Nongnat Tongkrajang[1], Porntida Kobpornchai[1,2], Pratima Dubey[1], Urai Chaisri[3], Kasem Kulkeaw◉[1,2]***

**1** Siriraj Integrative Center for Neglected Parasitic Diseases, Faculty of Medicine Siriraj Hospital, Mahidol University, Bangkok, Thailand, **2** Siriraj-Long Read Lab, Faculty of Medicine Siriraj Hospital, Mahidol University, Bangkok, Thailand, **3** Department of Pathology, Faculty of Tropical Medicine, Mahidol University, Bangkok, Thailand

* kasem.kuk@mahidol.edu

**Data Availability Statement:** All relevant data are in the manuscript and its supporting information files.

## Abstract

The lack of disease models adequately resembling human tissue has hindered our understanding of amoebic brain infection. Three-dimensional structured organoids provide a microenvironment similar to human tissue. This study demonstrates the use of cerebral organoids to model a rare brain infection caused by the highly lethal amoeba *Balamuthia mandrillaris*. Cerebral organoids were generated from human pluripotent stem cells and infected with clinically isolated *B. mandrillaris* trophozoites. Histological examination showed amoebic invasion and neuron damage following coculture with the trophozoites. The transcript profile suggested an alteration in neuron growth and a proinflammatory response. The release of intracellular proteins specific to neuronal bodies and astrocytes was detected at higher levels postinfection. The amoebicidal effect of the repurposed drug nitroxoline was examined using the human cerebral organoids. Overall, the use of human cerebral organoids was important for understanding the mechanism of amoeba pathogenicity, identify biomarkers for brain injury, and in the testing of a potential amoebicidal drug in a context similar to the human brain.

## Author summary

Brain inflammation caused by a free-living amoeba *Balamuthia mandrillaris* is really rare but life-threatening infectious disease. Given its rarity and difficulty to obtain a clinical isolate for further study, this disease has been neglected despite high mortality rate. Many gaps in our understanding of the disease remain opened. Emergence of the organoid platform allows modelling human diseases in an in vitro setting, leading to identification of potential molecular pathway as drug targets. Thus, this work attempts to deploy a human brain organoid to reveal such pathways, which are altered by the amoeba. A strain of *B. mandrillaris* was isolated from human biopsied brain. In the co-culture between brain organoid and amoeba, data suggested an alteration in neuron growth and increase of proinflammatory response. Brain organoid release brain trauma biomarkers. Importantly,

**Funding:** K. K. receives Specific League Funds from Mahidol University, Grant number (IO) R016521002; Siriraj Research Development Fund, Grant numbers (IO) R016433023, Faculty of Medicine Siriraj Hospital, Mahidol University; Siriraj Research Development Fund, Grant number (IO) R016637003, Faculty of Medicine Siriraj Hospital, Mahidol University; National Research Council of Thailand (NRCT): N42A650247. The funders had no role in study design, data collection and analysis, decision to publish, or preparation of the manuscript.

**Competing interests:** The authors have declared that no competing interests exist.

we show therapeutic effect of the antibiotic nitroxoline using the human cerebral organoids. Thereby, the use of the human cerebral organoids unravels the host response, identifies potential biomarkers, and provides an alternative way for testing a potential drug targeting the amoeba.

## Introduction

Some environmental protist species exert pathogenic effects upon entry into the human body. *Balamuthia mandrillaris* is an environment-dwelling amoeba that causes chronic inflammation in the brain, a condition termed *Balamuthia* amoebic encephalitis (BAE) [1]. As a free-living organism, *B. mandrillaris* has been isolated from soil [2,3], dust [4], water [5,6], and hot springs [7] worldwide. Regardless of the host's immune status, people of all ages are at risk of infection [8–12]. Although BAE is rare, the fatality rate was 90% among 109 infected individuals from 1974–2016 in the United States [13]. The true prevalence of *B. mandrillaris* infection is underestimated due to a lack of awareness and difficulty in differential diagnosis. Given the similar symptoms, BAE might be misdiagnosed as other brain infections caused by viruses, bacteria, or fungi that are more common [14]. Laboratory diagnosis is typically conducted postmortem via autopsy [10,13]. Definitive laboratory diagnosis primarily relies on immunodetection and histological examination of vegetative trophozoites or dormant cysts in the biopsied brain tissue, although the sensitivity is rather low [13,15]. Amplification of *Balamuthia* DNA is a feasible and highly sensitive method. Nevertheless, neither of these techniques is widely available. These factors lead to ineffective or delayed treatment, leading to unfavourable outcomes. The current treatment regimens rely primarily on various combinations of antifungals, antibiotics, and antiparasitics but are ineffective. However, on very rare occasions, these drug combinations can serve as radical cures [13,16–19]. Thus, an effective drug needs to be developed.

Disease models are an essential part of drug development and need to recapitulate the human physiological context [20]. To date, attempts to model brain infection have revealed mechanisms of host-pathogen interaction, including *Cryptococcus neoformans* and herpes simplex virus 1 [21,22]. The culture of primary human neurons is feasible [23] but challenging; the challenges include ethical concerns regarding the sources and accessibility of human brain tissue [24,25]. Neuroblastoma cell lines are widely available and easy to grow [22]. Nevertheless, two-dimensional cultures of human neuroblastoma cells differ drastically from the brain parenchyma due to a lack of cell heterogeneity and cell–cell or cell-matrix protein interactions. Thus, modelling human infectious diseases requires cell complexity and similarity to human tissue.

Recapitulation of human tissue in a miniaturized form is feasible owing to advances in stem cell culture [26]. Organoids can be generated from adult stem cells isolated from a given tissue or pluripotent stem cells [27]. In the presence of matrix proteins and growth factors, stem cells proliferate and differentiate into various cell types that are organized in a three-dimensional form and function like the tissue of origin or its counterpart. Compared to conventional 2D cultures, organoids are more physiologically similar to the counterpart tissue [28–31]. Human cerebral organoids are useful for the study of mechanisms underlying virus-caused microcephaly. By using human brain organoid, Zika virus impairs early stage of brain organoid development, resulting in smaller size. Treatment of the brain organoid with interferon-b could ameliorate the Zika-induced microcephaly [32]. Cerebral organoids can be generated from human induced pluripotent stem cells [33,34]. Therefore, there is less ethical concern

associated with the use of cerebral organoids compared to biopsies of brain tissue from human donors or even aborted foetuses. Herein, we used a coculture of human cerebral organoids with a clinical isolate of *B. mandrillaris* to observe cell damage. By this method, brain trauma-related biomarkers could be detected in the coculture medium. RNA sequence analysis suggested the occurrence of a proinflammatory response to the invading trophozoites. To identify a curative treatment for BAE [19], we demonstrate the use of cerebral organoids for testing the effect of the repurposed drug nitroxoline, an antibiotic for urinary tract infection, on trophozoites. Thus, cerebral organoids can serve as an alternative *in vitro* model for drug testing.

## Materials and methods

### Ethics statement

The Human Research Protection Unit, Faculty of Medicine Siriraj Hospital, approved the experiments, which included the use of human induced pluripotent stem cells (iPSCs) and a clinical isolate of *B. mandrillaris* (COA no. Si 146/2022).

### Culture of human iPSCs

The human iPSC line MUSli001-A was generated from skin fibroblasts from a caesarean section [35] and cultured on a polystyrene plate coated with 2 μg/mL Matrigel (Growth Factor Reduced; Corning, BD Bioscience) at 37°C with 5% $CO_2$. Essential 8 medium (Gibco) containing 100 IU/mL penicillin and 100 μg/mL streptomycin was used for cell growth and renewed daily. Cells were subcultured at a ratio of 1:10 every 5–6 days using 0.5 mM EDTA (Invitrogen) to detach clumped cells.

### Generation of human cerebral organoids

Cerebral organoids were generated following a stepwise protocol consisting of the formation of embryoids, differentiation of neuroepithelial cells, expansion of neural stem cells, and maturation of neurons [33]. The STEMdiff Cerebral Organoid Kit consists of basal media and supplements and was used following the manufacturer's protocol (STEMCELL, Canada). Embryoid bodies were formed using basal medium #1 mixed with supplement A. Then, 7-day embryoid bodies were embedded in Matrigel (Corning, Bedford, MA) and cultured in basal medium #1 mixed with supplement B for 5 days. To expand neural stem cells, Matrigel-embedded embryoid bodies were exposed to basal medium #2 mixed with supplements C and D for 7 days. The final step allowed the maturation of neurons in the organoid in basal medium 2 mixed with supplement E in an incubator with an orbital shaker (65–75 rpm) at 37°C with 5% $CO_2$. The neurons matured after culture in the final step for 60 days [36,37]. Maturity and cell identity were assessed based on the amplification of transcripts and immunodetection of proteins.

### Culture of *B. mandrillaris* trophozoites

Trophozoites of *B. mandrillaris* strain KM-20 were previously isolated from the biopsied brain tissue of a Thai patient diagnosed with *Balamuthia* amoebic encephalitis [38,39]. The minced brain tissue was initially cultured with human lung carcinoma A549 cells in DMEM supplemented with 10% FBS. In the present study, the KM-20 strain was grown in human cell-free culture using BM-3 medium, which consisted of peptone (4 mg/mL), yeast extract (4 mg/mL), yeast RNA (1 mg/mL), ox liver digest (10 mg/mL), lipid mixture (1 mg/mL), hemin (2 μg/mL), taurine (50 ng/mL) and vitamin mixture in Hanks' balanced salt [40]. The trophozoites were routinely subcultured every 8 to 10 days. Trophozoites were transferred into 2 mL of BM-3

medium in a 6-well plate in a volumetric ratio of 1:5. The BM-3-cultured trophozoites were used for all the experiments.

## Amoebicidal activity

The half-maximal inhibitory concentration (IC50) of nitroxoline and miltefosine was calculated as described in a previous report [41]. A 10 mM stock of each drug was dissolved in DMSO. The BM-3-derived trophozoites were added to 96-well plates at a density of 8,000 cells per well in 50 µL of BM-3 medium. Twofold serial dilutions of the drugs were prepared in BM-3 medium, yielding final concentrations of 0.78–4400 µM and 0.078–440 µM for miltefosine and nitroxoline (obtained from Dr. Christopher A Rice at Purdue University College of Veterinary Medicine), respectively. Control groups were treated with the vehicle (DMSO), with the treatment concentration corresponding to the final concentration of DMSO in each drug dilution. The morphology of the drug-exposed trophozoites was imaged after 72 hours. Following drug exposure for 72 hours, 50 µL of CellTiter Glo reagent (Promega, Madison, WI) was added to each well. Luminescence was measured using a multidetection microplate reader (Biotek, Synergy H1). The viability of *B. mandrillaris* was calculated using the following formula: (luminescent signal of the drug treatment group/luminescent signal of vehicle control group) x 100. The IC50 was calculated using a dose–response curve in GraphPad Prism ver 6.0. In the coculture experiments, the mature brain organoids were inoculated with *B. mandrillaris* trophozoites in 24-well plates containing brain culture medium. Each plate was incubated at 37°C and 5% $CO_2$ for 1 hour to ensure a close distance between the trophozoites and brain organoids before treatment with 100 µM miltefosine, 35 µM nitroxoline, or vehicle control. The human brain organoid alone was used as a negative control for this experiment. At 20 hours postdrug exposure, the medium was removed and replaced with fresh medium. After incubation for 4 days, the sample cells were collected for preparing paraffin-embedded tissue sections. Characteristic morphological changes in brain organoids were investigated daily under an inverted microscope.

## Coculture of *B. mandrillaris* with human cerebral organoids

The trophozoites were cultured with 60-day-old organoids using the cerebral organoid maturation medium. The cells were incubated at 37°C in a humidified atmosphere containing 5% $CO_2$. The culture medium was changed every 3–4 days. The collected medium was stored at -20°C before ELISA. After 16 days of coculture, the cerebral organoids were collected for further analysis.

## Pathological observation

For histological examination, cerebral organoids were fixed in 4% paraformaldehyde at RT for an hour. After two washes with PBS, the samples were placed in a tissue processor (Milestone, Italy), and each piece of tissue was embedded in a paraffin block. A 4-µm thick tissue slide was prepared using a microtome, laid down on a coated glass slide, and subsequently stained with haematoxylin and eosin (H&E). These sections were examined under a light microscope (Olympus, model BX41), and the images were captured and saved by the DP2-Bsw Program. Cell and tissue types in cerebral organoids were identified by a pathologist.

## Transmission electron microscopy (TEM)

Transmission electron microscopy (TEM) was performed according to previous reports [42,43]. Briefly, the organoids were fixed with 2.5% glutaraldehyde in 5% sucrose phosphate

buffer (pH 7.4) for an hour. After three washes with phosphate buffer, the fixed organoids were incubated with 1% osmium tetroxide for an hour and dehydrated in an ethanol gradient. Following infiltration, the organoids were embedded in epoxy resin and subjected to sectioning using an ultramicrotome (Leica UC&, Germany). Sections were mounted on copper grids (200-mesh square) and stained with 2% uranyl acetate and lead citrate solution before examination under a transmission electron microscope (HT7700, Hitachi Global Company, Japan) at 100 kV with 2.0k magnification.

## RNA sequencing

Organoids were collected at 16 days postcoculture with the trophozoites. For the coculture and control experiments, three biological replicates of cerebral organoids were subjected to RNA sequencing. Total RNA was extracted using a PureLink RNA Mini Kit (Thermo Fisher Scientific, MA). The RNA sequencing workflow included five main steps. The quality and quantity of total RNA were examined using an Agilent TapeStation (CELEMICS) and analysed using STAR 2.7.10b. Following cDNA synthesis from the RNA samples, library preparation was performed to carry out NGS sequencing on the Illumina platform. Bioinformatics analysis was performed for data QC, expression level profiling, and variant/gene fusion detection. For data analysis, differential expression analysis was performed using featureCounts v.2.0.3. Gene ontology enrichment analysis was performed using the Gene Ontology Consortium and PANTHER classification system.

## ELISA

To assess early cell damage in the human cerebral organoids, the culture medium was collected at days 3, 7, and 11 postcoculture with the trophozoites. Ubiquitin carboxyl-terminal hydrolase family 1 (UCH-L1) is an indicator of damage to the neuronal cell body, while glial fibrillary acidic protein (GFAP) is an indicator of astroglial injury [44–46]. ELISA was performed according to the manufacturer's instructions. The ELISA kit for UCH-L1 was from Thermo Fisher Scientific (Frederick, MA), and that for GFAP was from MA Abcam (Cambridge, UK).

## Quantitative gene expression analysis

RNA was extracted using a PureLink RNA Mini Kit (Thermo Fisher Scientific, MA) and subjected to cDNA synthesis using an iScript Reverse Transcription Supermix cDNA Synthesis Kit (Bio-Rad, CA). The expression levels of cell-specific transcripts were examined using quantitative reverse-transcription PCR. Table 1 shows primer sets obtained from previous reports [33]. Amplification of cDNA was performed using Luna Universal qPCR Master Mix (New England Biolabs, MA) and 10 μM each primer (Table 1). After a 3-min denaturation at 95˚C, DNA amplification was performed with 40 rounds of DNA denaturation at 95˚C for 15 seconds and primer annealing and extension at 60˚C for 30 seconds. Transcripts of actin beta (*ACTB*) or glyceraldehyde-3-phosphate dehydrogenase (*GAPDH*) served as internal controls for normalizing the level of gene expression. The threshold cycles of each transcript were compared using the $2^{-\text{delta delta CT}}$ method [47]. The level of gene expression is expressed as the relative expression among the samples. Three independent experiments were subjected to quantitative reverse-transcription PCR, which was carried out in triplicate for each run.

## Statistical analysis

All data and graph generation were conducted using GraphPad Prism software version 6.0 (GraphPad Software, Inc., San Diego, CA, USA). The parametric methods of student t-test and

**Table 1. Primer sequences used for quantitative PCR in the study.**

| Gene | Primer (5'-3') |
|---|---|
| OCT4A | Fw: GGAGAAGCTGGAGCAAAACC |
|  | Rv: TGGCTGAATACCTTCCCAAA |
| NANOG | Fw: GATTTGTGGGCCTGAAGAAA |
|  | Rv: CTTTGGGACTGGTGGAAGAA |
| SOX1 | Fw: TATCTTCTGCTCCGGCTGTT |
|  | Rv: GGGTCTTCCCTTCCTCCTC |
| PAX6 | Fw: AGTTCTTCGCAACCTGGCTA |
|  | Rv: ATTCTCTCCCCCTCCTTCCT |
| FOXG1 | Fw: AGGAGGGCGAGAAGAAGAAC |
|  | Rv: TGAACTCGTAGATGCCGTTG |
| SIX3 | Fw: CTATCAACAACCCCCAACCA |
|  | Rv: AGCCGTGCTTGTCCTAGAAA |
| KROX20 | Fw: TTGACCAGATGAACGGAGTG |
|  | Rv: CTTGCCCATGTAAGTGAAGGT |
| ACTB | Fw: AAATCTGGCACCACACCTTC |
|  | Rv: AGAGGCGTACAGGGATAGCA |
| GAPDH | Fw: GCATCCTGGGCTACACTGAG |
|  | Rv: TGCTGTAGCCAAATTCGTTG |

one-way ANOVA were done to determine the statistical differences between two or more than three groups. The results are presented as means ± standard deviation (SD). The *P*-value <0.05 was considered statistically significant.

## Results

### Human iPSC-derived cerebral organoids contain mature neurons and are functional

Following the schematic diagram shown in Fig 1A, on day 5 of cell differentiation in a low-attachment, U-shaped well, the outer area of the embryoid body was translucent, while the centre was a dark necrotic zone. On day 10, dome-tipped rod shapes projected from the centre of the Matrigel-embedded embryoid body, indicating the expansion of neuroepithelial cells. From day 10 onwards, the protruding dome structure became spherical (Fig 1B). To examine neuron specification and maturation, the levels of neuron-specific transcripts were compared at days 10, 40, and 60 post-iPSC differentiation. The levels of the pluripotency markers *OCT4A* and *NANOG* decreased during organoid differentiation. Neurogenesis, indicated by *SOX1* expression, increased when the neuroepithelial bud was exposed to the culture medium used for neuron maturation. On day 60, the level of the *SOX1* transcript decreased, while that of radial glial cell-specific *PAX6* increased. To confirm early brain regionalization in the 3D cell structure, the expression levels of *FOXG1* and *SIX3* (forebrain markers) and those of *KROX20* and *ISL1* (hindbrain markers) were examined. FOXG1 is a transcription factor essential for forebrain development that is upregulated starting from the expansion of neuroepithelial cells to the maturation of neurons. SIX3 is reportedly expressed only in the anterior neuron plate of the mammalian embryo (developing eyes, pituitary gland, hypothalamus, and olfactory placodes) [48]. The *SIX3* transcript level decreased from day 40 onwards. In contrast, the *FOXG1* transcript level increased on day 40 and decreased by day 60. EGR2 expression increased on day 60 in the 3D cell structure, indicating early hindbrain development (Fig 1C).

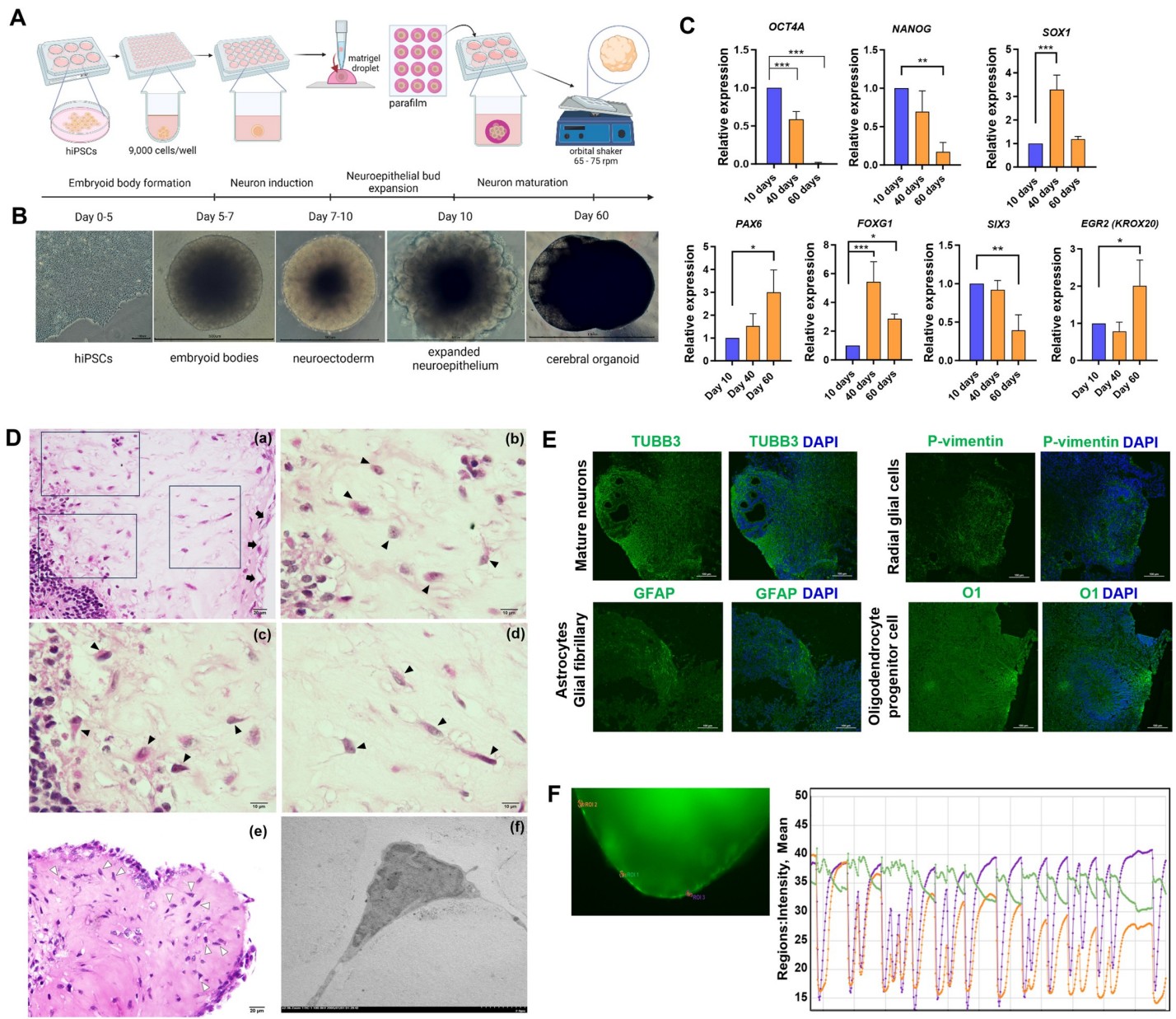

**Fig 1. Human cerebral organoids possess key characteristics of the adult brain.** (A) Schematic diagram of cerebral organoid generation from human induced pluripotent stem cells (hiPSCs). The key experiments consisted of embryoid body formation, neuron induction, neuroepithelial bud expansion, and neuron maturation. From day 10 onwards, the Matrigel-embedded organoids were orbitally shaken to allow the perfusion of oxygen and nutrients into the organoid. Created with Biorender. (B) Representative microscopy images of hiPSCs, embryoid bodies, neuroectoderms, neuroepithelia, and cerebral organoids (scale bars, 0.5–1 mm). (C) Expression levels of genes functioning in pluripotency and neurogenesis. Transcript levels are reported relative to the 10-day sample. The data are the mean of the relative expression level ± SD. (D) H&E-stained cerebral organoids were cultured and examined under a light microscope. Neuroblasts or primitive nerve cells were under a meninge-like layer (black arrows in subpanel a; 400X). The rectangles are areas displayed at higher magnification in subpanels b-d. The specific characteristics of neuroblasts are a larger size than that of a mature nerve cell and an euchromatic nucleus with a prominent nucleolus (block arrowheads in subpanels b, c, and d (1,000X). The typical shape of a pyramidal cell (mature neuron) is indicated by white arrowheads (e; 400X). A representative image of mature neurons observed using a transmission electron microscope (f; 400X). (E) Immunofluorescence of neurons (TUBB3+), radial glial cells (P-vimentin+), astrocytes (GFAP+) and oligodendrocyte progenitor cells (O1+) in cerebral organoids (scale bars, 100 mm). (F) Calcium influx and efflux of the cerebral organoids using Fluo-4 and live imaging for 20 minutes (regions of interest (ROIs) are outlined in the left panel) as measured by changes in fluorescence (arbitrary units). Three individual organoids were tested (n = 3).

On day 60, neuroblasts and mature neurons were observed under the meninge-like structure based on H&E staining. Meninges and ependymal cells support nerve tissue. A single layer of squamous epithelial lining-like meninges covers the 3D cell structure as a capsule, as shown by arrowheads in Fig 1D (a). The specific character of neuroblasts was a large, euchromatic nucleus with a prominent nucleolus observed with higher magnification (arrowheads in subpanel b-d, Fig 1D). Notably, a pyramidal shape is typical for mature neurons in the cerebral cortex, and many pyramidal cells were observed (white arrowheads in subpanel e, Fig 1D). Moreover, under TEM, the triangular shape of the nucleus and delicate cytoplasm with elongate axons and dendrites observed were indicative of mature neurons, as shown in Fig 1D (f).

Next, we examined the presence and organization of neurons, astrocytes, and radial glial cells in the 60-day 3D cell structure using cell-specific antibodies. Based on the TUBB3 protein, mature neurons were widely dispersed throughout the section of the 3D cell cluster (Fig 1E). In contrast, P-vimentin-expressing radial glial cells and glial fibrillary astrocytes were observed in some areas. Similar to the pattern of TUBB3-positive neurons, oligodendrocyte progenitor cells were distributed in a wide area. Thus, the various patterns of cell distribution indicated distinct types and specific rearrangements of each cell population. To test the function of neurons in calcium transport across the cell membrane, the influx and efflux of calcium were monitored in real time. After an hour of incubation and subsequent removal of excess $Ca^{2+}$-binding Fluo-4, the green fluorescence of the 3D cell clusters was captured at 15-second intervals for a total of 20 minutes, yielding 1,199 images. The intensity of fluorescence was plotted on the Y-axis with time on the X-axis (Fig 1F). Multiple peaks were observed for Fluo-4, suggesting the influx and efflux of calcium. Overall, the spherical-shaped 3D cell cluster possessed critical characteristics of the brain parenchyma.

## Morphological changes in neurons in human cerebral organoids cocultured with *B. mandrillaris* trophozoites

Previous studies have demonstrated that 60-day cerebral organoids are representative of an adult human brain [36,37]. Thus, 60-day cerebral organoids were cocultured with *B. mandrillaris* trophozoites that were regularly maintained in host-free BM-3 medium. A period of 16 days of coculture was selected because trophozoites were observed inside the organoids together with hollow and dead cells at this time point (S1 Fig). After an hour of coculture, we examined the spatial distribution of the trophozoites using differential fluorescence labelling. 3D imaging (Z-stack) confirmed that the trophozoites adhered to the outermost layer and invaded the organoid (magenta in S2A Fig; S1 and S2 Videos). On day 11 of coculture, snapshot imaging showed the attachment of the trophozoites on the cerebral organoid (S2B Fig). At the outermost layer, the trophozoites remained active in the protruding cytoplasm (S3 Video). Multiple elongated branched structures were observed, while one side adhered to the organoid (arrowhead in left panel, S2B Fig). In the deeper layer, some trophozoites became round or oval and exhibited a less elongated cytoplasm (arrowheads in middle panel, S2B Fig). Notably, we observed a double-wall morphology, which resembled that of a cyst (arrow in right panel, S2B Fig).

Following 16 days of coculture, H&E staining revealed that all the *B. mandrillaris* cells were in the trophozoite stage (Fig 2A). The trophozoites were located inside and on the outermost surface of the organoids (Fig 2A). At higher magnification, the nuclei of the neurons were found to have lost the pyramidal shape (arrows in subpanels c and d, Fig 2A). Inside the cerebral organoids, the trophozoites were polymorphic (subpanel c of Fig 2A). In contrast, the outermost trophozoites were mostly round (asterisks in subpanels a and b, Fig 2A). As shown within the dotted line of subpanel d of Fig 2A, neuronal nuclei of various sizes (arrows) were

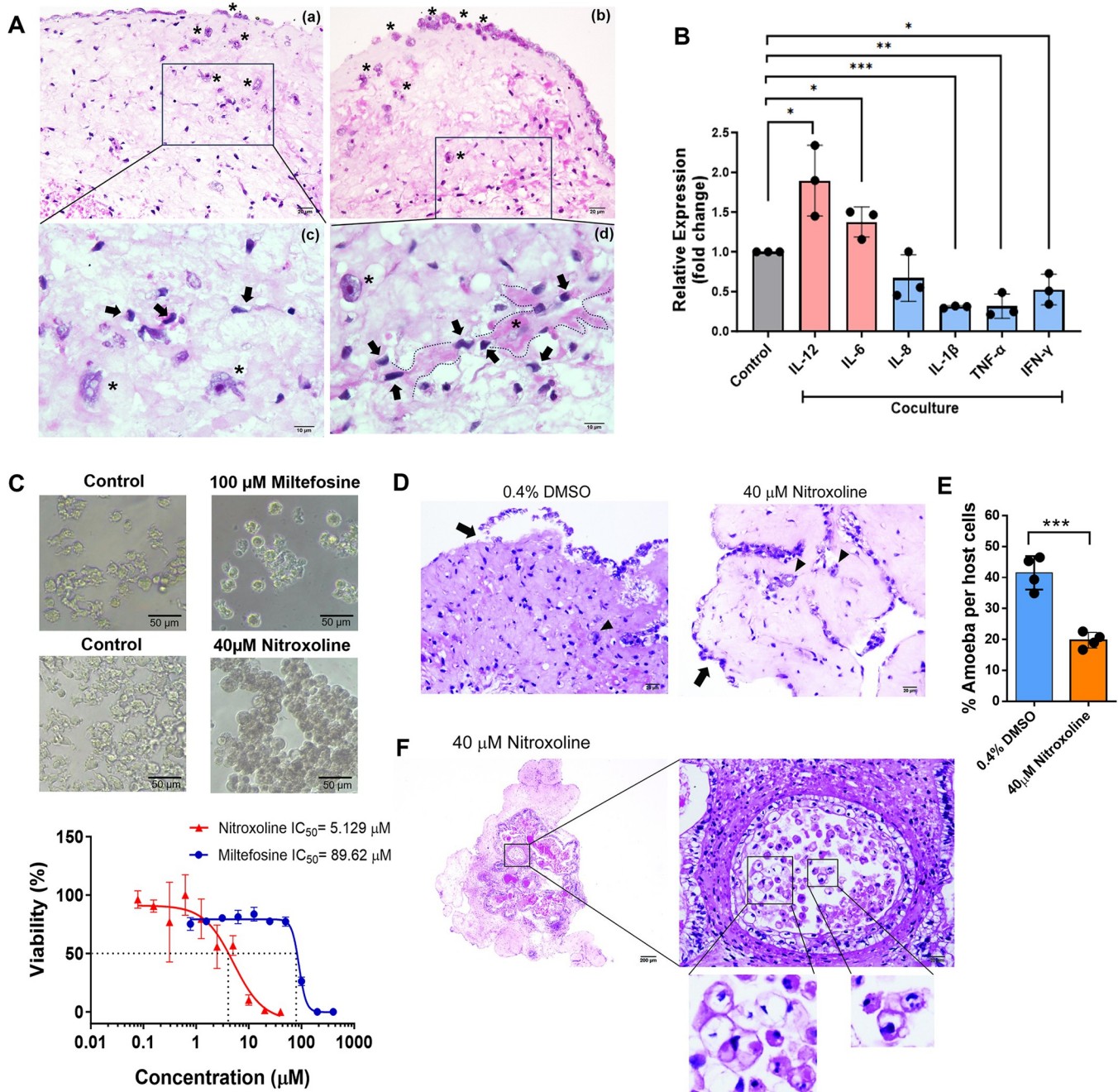

**Fig 2. Cytopathic effect of *B. mandrillaris* against human cerebral organoids.** (A) H&E staining of the coculture of *B. mandrillaris* with cerebral organoids. *B. mandrillaris* trophozoites first attached to a meninge-like layer (asterisk) (subpanel a and b; scale bars, 20 um), a single-layer squamous epithelium covering the cerebral organoids as a capsule. As shown in the zoomed-in images of the insets in the black box, *B. mandrillaris* trophozoites invaded the cerebral organoids (asterisks in subpanels c and d; scale bars, 10 μm). As indicated by the dotted line, a trophozoite projected long pseudopodia proximal to neurons that were degenerated and necrotic (arrows in subpanel c and d; scale bar, 10 μm). (B) Expression levels of human genes encoding proinflammatory, inflammatory and immunoregulatory cytokines after 16 days of coculture of cerebral organoids with trophozoites. The data are the mean of the relative expression level ± SD. The statistical analysis was performed by one-way ANOVA, and the significance of data are shown as *$p < 0.05$, **$p < 0.01$, and ***$p < 0.001$. (C) The dose–response curve of nitroxoline and milefosine. The effect of nitroxoline and milefosine were examined based on the viability of *B. mandrillaris* trophozoites at 72 hours postexposure. Representative microscopy images of *B. mandrillaris* trophozoites after treatment with 0.4% DMSO as a vehicle control, 40 μM nitroxoline and 100 μM milefosine. Data from triplicate experiments were used for calculation of the half maximal inhibitory concentration (IC50). (D) H&E staining of cerebral organoids cocultured with trophozoites with and without nitroxoline (scale bars, 20 μm). In the DMSO-treated control (left panel), the trophozoites invaded the inner layer (arrowhead), while the meninge-like layer detached from the outermost layer (arrow). In the nitroxoline-treated samples (right panel), the outermost layer of the cerebral organoids remained intact (arrow). Some trophozoites had a large unstained

space in the cytoplasm (arrowheads). (E) Bar graph showing the number of amoeba per 100 human cells from three representative microscopic fields (magnification of 100X, objective lens) of H&E-stained sections. Four independent examiners counted the number without knowing the sample types to avoid bias. (F) Trophozoites in the hollow inside the cerebral organoids after nitroxoline exposure. The hollow is indicated in the lower magnified image of the cerebral organoids (left panel with rectangle). The higher-magnification image of the hollow shows numerous trophozoites (scale bar, 20 μm). Most of the trophozoites had fragmented nuclei and large empty spaces in their cytoplasm (the zoomed-in images of the insets in the black boxes of the right panel).

observed in the area proximal to the trophozoites (asterisk). However, there is no difference in the size of the cerebral organoids between the control and coculture with *B. mandrillaris* ($p<0.05$) (S3 Fig). Next, we examined the host response to the trophozoites using gene expression analysis. Relative to that in noninfected cerebral organoids, the transcript levels of the proinflammatory markers IL-12 and IL-6 were significantly upregulated in the infected organoids. None of the chemokine (IL-8), inflammatory regulator (IL-1β and TNF-α), or immunostimulatory IFN-γ transcript levels increased in the coculture. Taken together, these data support the pathological findings for the human brain infected with *B. mandrillaris* and point to neuron preference.

## Nitroxoline ameliorates cellular damage in human cerebral organoids

Nitroxoline was recently identified as a potent amoebicidal drug against *B. mandrillaris* [41] and shown to be a curative treatment in a BAE patient [19]. It remains unknown whether the *B. mandrillaris* strain KM-20 is susceptible to nitroxoline. Therefore, we first determined the amoebicidal effect of miltefosine and nitroxoline against *B. mandrillaris* KM-20. After 72 hours of drug exposure, the trophozoites exhibited a round morphology without cytoplasmic protrusion compared to the DMSO-treated control (Fig 2C). Based on the ATP level, the IC50 values of miltefosine and nitroxoline were 89.6 and 5.1 μM, respectively (Fig 2C). Thus, a dose of 6xIC50 was used in the drug test with cerebral organoids. The previous study suggests the effect of nitroxoline on the delayed destruction of human brain tissue. Thus, we confirmed the protective effect of nitroxoline by following the study of Laurie et al [41]. The cocultures of brain organoids with *B. mandrillaris* trophozoites were concurrently exposed to nitroxoline and the brain organoid damage was assessed on day 4 post-drug exposure. In the absence of nitroxoline, the trophozoites invaded the inner layer (arrowhead in left panel, Fig 2D), while the meninge-like layer detached from the outermost layer of the cerebral organoids (arrow in left panel, Fig 2D). In contrast, the outermost layer of the treated cerebral organoids remained intact (arrow in right panel, Fig 2D). Some trophozoites had a large unstained space in the cytoplasm (arrowheads in right panel, Fig 2D). To quantify the nitroxoline effect on infection, the number of *B. mandrillaris* is enumerated based on polymorphic and round shape, while that of other cells is counted based on the nucleus stain (S4 Fig). The treated cerebral organoids showed a considerable reduction in the percentage of *B. mandrillaris* ($p<0.001$) (Fig 2E). Notably, there were many trophozoites located in the hollow region inside the cerebral organoids. A multilayer of cells with round or oval-shaped nuclei surrounded the hollow region. Some trophozoites had fragmented nuclei or large unstained areas in the cytoplasm (magnified views of the rectangles in Fig 2F). However, there were no trophozoites located in the hollow region of the brain organoids that were exposed to DMSO as vehicle control (S5 Fig). For miltefosine, the vacuolated trophozoites were not presented in the hollow or inner layer of the cerebral organoid. Thus, it seems that the specific to the nitroxoline treatment (S5 Fig). Thus, nitroxoline exhibits cytotoxicity against trophozoites and ameliorates cellular damage in human organoids.

## Axon formation and ECM organization are likely amended in human cerebral organoids cocultured with *B. mandrillaris*

Next, we examined transcriptional changes by performing bulk RNA sequencing. Statistical analysis of up- and down-regulated genes and the list of top 100 highly expressed genes are shown in the S1 and S2 Tables, respectively. First, we confirmed the identity of the human cerebral organoids. The data set of RNA sequences was compared with other data sets. A heatmap of RNA sequences shows that the 60-day cerebral organoids were similar to other brain organoids but differed from other tissues, including smooth muscle, adipocytes, retinoblastoma, cardiac myocytes, bronchial epithelial cells, and placenta (Fig 3A). Three biological replicates of the 16-day coculture experiments were performed to represent the individual brain and observe the intersample variation. Based on the heatmap in Fig 3B, the gene expression levels differed among the three sets of noncoculture experiments (control). Despite high variation, upregulated and downregulated genes were observed. These genes were involved in several cellular pathways with different degrees of significance, as indicated by P values (dark orange to yellow represent high to low, Fig 3C). The most significant change was closely related to axon guidance, in which neurons project their nerve fibres to reach the targets. The expression of genes functioning in organizing the extracellular matrix and haemostasis decreased (Fig 3C). As shown in the volcano plot, the upregulated transcripts were *INHBE* and *CCL7*, while the downregulated transcripts were *ACTC1* and *TKTL1* (Fig 3D). Gene ontology functional analysis showed pathways in which significant changes were detected in the coculture with *B. mandrillaris*. The KEGG pathways included amoebiasis and AGE-RAGE signalling pathways in diabetic complications. Fibrillar collagen trimer and collagen fibril organization were present among the cellular component and biological process terms. Notably, oxidoreductase and NADH dehydrogenase transcripts were absent in the coculture experiment (Fig 3E). Overall, the main defect was axon formation of neurons accompanied by alteration in ECM organization.

## UHC-L1 and GFAP are surrogate markers of neuron damage

CCL7 is released from astrocytes to induce microglia-mediated proinflammation in mice [49] and rats [50] with traumatic brain injury. Next, we examined the release of two biomarkers of traumatic brain injury, neuron-specific UCHL-1 and astrocyte-specific GFAP, in the culture medium [51]. UHC-L1 was detected in the culture medium on day 3 postcoculture with *B. mandrillaris*. The level of UHC-L1 significantly increased more than 10-fold compared to that in the noninfected control ($p<0.001$) (Fig 4A). On day 7, the level of UHC-L1 in the coculture was the highest and was still greater than that in the control experiment. On day 11 post coculture, the UHC-L1 level gradually decreased to a level similar to that on day 3. Similar to the UHC-L1 level, the level of GFAP increased incrementally from day 3 to day 7 postcoculture. Despite a slight decrease on day 11, the level of GFAP in the coculture was still higher than that in the control experiment (Fig 4B). There were no significant changes in UHC-L1 and GFAP in the culture medium of the noninfected control. The ratio between the GFAP and UCH-L1 concentrations was used as an index of traumatic brain injury [51]. The glial neuronal ratio was calculated as the GFAP concentration (ng/mL) divided by the UCH-L1 concentration (ng/mL). The glial neuronal ratios were 10, 16, and 30 on days 3, 7, and 11 postcoculture, respectively (Fig 4C). Compared to the GNR proposed by Mondello et al. [51], this finding suggested the existence of a focal mass lesion.

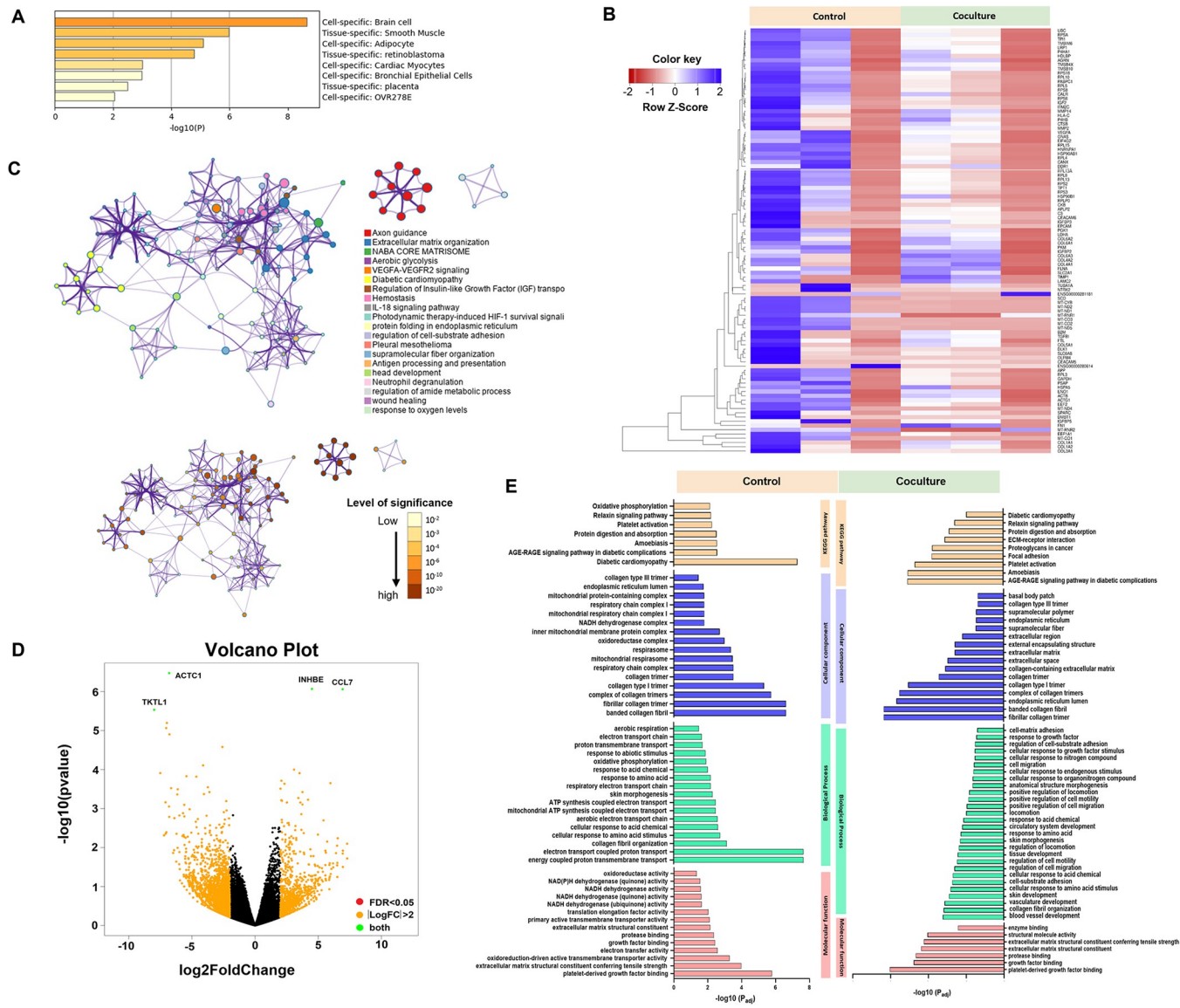

**Fig 3. The transcription profile of human cerebral organoids in coculture with *B. mandrillaris* trophozoites.** (A) Identification of cell types in the 60-day cerebral organoids based on the types and levels of all detected transcripts. (B) Heatmap of the transcripts detected in the cerebral organoids after 16 days of culture with the trophozoites. Upregulated and downregulated transcripts are highlighted. There are six samples (control = 3; test = 3). (C) Gene Ontology (GO) of the downregulated transcripts in the cocultured cerebral organoids. GO bar chart. The upper panel shows each cluster of genes in different ontogeny with different colors, while the lower panel indicates a "low to high level" of statistically significant differences as light yellow to dark red. (D) Volcano plot of the differentially expressed genes (DEG) between the control and coculture groups. The graph showing log2 fold change (x-axis) against −log10 p-value (y-axis) of transcripts identified by RNA sequence analysis. The up- and down-regulated expression datasets that meet the criterion for log2 FC > 2 are presented in yellow dots. The transcripts that are identified as significantly differentially expressed (*p*-value < 0.05) are indicated in red. The green dots are highlighted as both the criterion for log2 FC > 2 and significantly different transcripts, while the black dots indicate non-significant transcripts. (E) Classification of the transcripts based on KEGG pathways, biological processes, cellular responses and molecular functions.

## Discussion

Organoids have become useful *in vitro* disease models, particularly for rare diseases and diseases that are difficult to model using conventional systems. Here, we demonstrate the use of human cerebral organoids to model an amoebic brain infection, which is very rare and fatal. A coculture of cerebral organoids with a clinical isolate of *B. mandrillaris* strain KM-20 revealed

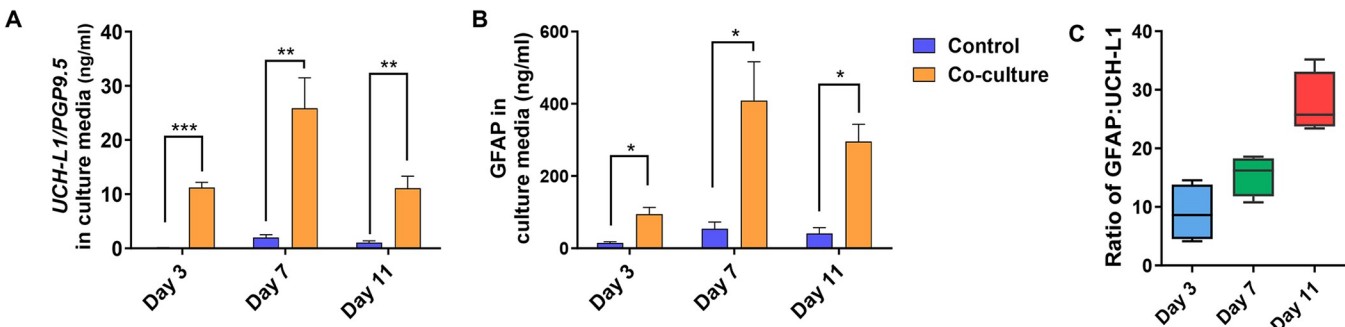

**Fig 4. Brain trauma biomarkers were elevated in the culture medium of human cerebral organoids cultured with _B. mandrillaris_.** (A, B) Levels of human GFAP and UCH-L1 in the culture medium of cells harvested at days 3, 7, and 11 postcoculture with _B. mandrillaris_ trophozoites. Two independent cerebral organoids were examined. The data are the mean ± SD. The statistical analysis was performed by an independent sample t test, and the significance of the data are shown as $*p<0.05$, $**p<0.01$, and $***p<0.001$. In both panels, black bars = control; grey bars = coculture. (C) The ratio of GFAP and UCH-L1 in the culture medium of noninfected and infected cerebral organoids (n = 2).

host responses at the transcriptional level, by which the discovery of two potential biomarkers of brain damage could be guided. The amoebicidal effect of nitroxoline could be examined using a human cerebral organoid alternative to brain tissue explants.

In early 2023, a rare successful treatment of BAE was reported [19]. A patient survived after receiving a combination of drugs, including nitroxoline, an antibiotic for the treatment of urinary tract infections. As a repurposed drug, nitroxoline possesses amoebicidal activity against cysts and trophozoites. The IC50 of nitroxoline against _B. mandrillaris_ strain V451, which was isolated from the human brain and established as a cell line by coculture with Vero cells, was slightly lower than that found against the KM-20 strain used in this study (2.84 and 5.13 μM, respectively). The difference might be due to differences in the strain and methods for maintenance of the trophozoites. Following exposure to 35 μM nitroxoline, human primary brain tissue cocultured with _B. mandrillaris_ V451 remains intact without gross tissue damage [41]. Consistent with this report, there was no damage to human cerebral organoids that were cocultured with _B. mandrillaris_ KM-20 after exposure to nitroxoline. Thus, human cerebral organoids can be used in _in vitro_ drug testing to replace primary brain tissue, the availability of which is limited. Nevertheless, the drug administration performed in this study differs from the clinical setting of BAE. The drug regimen is administrated into the BAE patients after the infection is already established. Simultaneous treatment of the _B. mandrillaris_-infected brain organoid aims to confirm the protective effect of nitroxoline as described by Laurie et al [41]. This limitation of the experimental design can be further addressed in the context of BAE patients, in which the coculture of brain organoids with _B. mandrillaris_ will be first set up until cells are damaged and then followed by drug exposure.

Transcript profiles allowed us to identify neurons as a major target and identify the changes that occurred at the transcription level when cerebral organoids were infected with _B. mandrillaris_ trophozoites. Among the downregulated genes, ACTC1 is an actin involved in the migration of astrocyte-derived glioblastoma cells [52], while TKTL1 functions in the generation of neurons [53]. Among the upregulated genes, INHBE is a subunit of inhibin, which is an antagonist of activin signalling. Activin reportedly promotes neuron growth and axon guidance in Drosophila [54,55]. In addition, CCL7 is upregulated in the cerebral organoids co-cultured with _B. mandrillaris_ and reportedly released from astrocytes and mediates neuroinflammation via activation of microglia. Under experimental conditions, CCL7 activated microglia in a traumatic brain injury model in mice [49] and rats [50]. Thus, neuron generation is altered, while

the activated astrocyte microenvironment is favourable for inflammation. For accurate interpretation, the gene expression levels must be further validated by examining protein levels.

A cerebral organoid is useful to study immune responses, particularly in Zika-induced microcephaly [32] IL-8, IL-1β, TNF-α, and IFN-γ reportedly induce intestinal inflammation caused by *Entamoeba histolytica*. They recruit neutrophils and macrophages to the site of amoebic invasion. Thus, a decrease of IL-8, IL-1β, TNF-α, and IFN-γ likely allows the amoeba to escape from inflammatory immune cells [56]. Thus, reduction of IL-8, IL-1β, TNF-α, and IFN-γ imply the microenvironment devoid of immune responses, resulting in the survival of the *B. mandrillaris*. However, it needs to confirm cytokine release at the protein level and functional tests to conclude.

Traumatic brain injury can be identified based on the release of neuron-specific proteins. UCH-L1 is a neuron-specific protein that hydrolyses C-terminal adducts of ubiquitin to generate ubiquitin monomers. It is present in neurons and nerve fibres in the central and peripheral nervous systems. UCH-L1 is a useful marker of neurodegenerative disorders and brain trauma. Glial fibrillary acidic protein (GFAP) is a class-III intermediate filament, a cell-specific marker that distinguishes astrocytes from other glial cells during the development of the central nervous system. A meta-analysis reported a significant increase in serum UCH-L1 levels in patients with traumatic brain injury compared to controls [44]. GFAP and UCH-L1 have been used as biomarkers of traumatic brain injury [45]. Studies have reported an increase in blood GFAP levels in neuroinflammation with potential clinical applications [46]. In cerebral spinal fluid and serum, the levels of GFAP were significantly higher in individuals with West Nile virus infection than in controls; these findings were correlated with the severity of postmortem histopathology [57]. Further validation with clinical samples from BAE patients may highlight utility of these biomarkers for early diagnosis and for studying response to treatment. For instance, reduction in levels of the brain damage markers should be investigated in the nitroxoline-treated cerebral organoids. Nevertheless, the scarcity of BAE has hindered such clinical validation.

One of the limitations of the use of cerebral organoids is the heterogeneity of cell types. The initial phase is the formation of embryoid bodies, consisting of the ectoderm, mesoderm, and endoderm. The subsequent phase is to induce ectoderm to become neuroepithelial cells, which form column shapes budding from the outer surface of the embryoid body. These neuroepithelial buds have fluid-filled lumens. The culture medium for neural induction contains a minimal medium that does not support the development of the endoderm and mesoderm. Thus, it is unlikely that the mesoderm and endoderm located inside the embryoid body would develop. However, it is suggested that 30–80% of organoids in Matrigel contain neuroepithelial buds. Using single-cell analysis of 31 organoids, Quadrato et al. reported that there was variation in cell type proportions across different batches. All the analysed batches contained mesoderm-derived cells, such as muscle cells, in the human cerebral organoids [58]. Consistent with this, we also found genes related to smooth muscle following the brain-specific genes. Cerebral organoids grown in the same spinning bioreactor were identical in terms of cell types. Thus, the environment in a spinning bioreactor possibly regulates cell identity in cerebral organoids. Here, we used an orbital shaker at the final step of neuron maturation, which likely explains the cell heterogeneity that we observed. Such variation occurs even in the same batch of experiments. Several factors have been proposed as causes, including pluripotency of iPSCs at the starting point of EB formation, the timing of neural induction, and embedding in Matrigel [33,34].

Cerebral organoids are more suitable for drug testing following a high-throughput screening process [41]. The cost of generating cerebral organoids is higher than that of 2D culture of iPSC-derived neurons. Generating cerebral organoids requires a long period of neuron

maturation. Cell analysis remains complicated, requiring sectioning into a thin layer of cells followed by immunofluorescence detection. To further improve the organoid platform for BAE, the addition of immune cells would allow the study of granulomatous encephalitis. Pathological examinations of biopsied brain tissues reveal infiltrating lymphocytes with necrotic cells [39]. Microglia have been shown to autonomously develop within organoids [59]. Alternatively, microglial precursor cells can be cocultured with cerebral organoids to allow the integration of microglia [60,61]. Nevertheless, the integration of adaptive immune cells in organoids remains unexplored.

## Supporting information

**S1 Fig. Cytopathologic effect of *B. mandrillaris* trophozoites on human cerebral organoids.** A 16-day coculture of *B. mandrillaris* trophozoites with human cerebral organoids was subjected to H&E staining. (A) Trophozoites of *B. mandrillaris* were observed inside the cerebral organoid, in which the cells formed clumps (white insets). (B and C) Higher magnification of insets in panel A shows trophozoites in the cell clump in the upper and lower inset of panel A, respectively. (D and E) Zoomed-in images of the insets in panels B and C, respectively. Yellow arrows indicate trophozoites with a round shape with nuclei in some cells. F. Noninfected cerebral organoids showed well-defined nuclei without cell clumps. Scale bars in panels A-C = 100 μm; scale bars in panels D-E = 50 μm.
(TIF)

**S2 Fig. Coculture of *B. mandrillaris* trophozoites with human cerebral organoids.** (A) Representative fluorescence images of cerebral organoids cocultured with trophozoites (scale bars, 200 μm). The cerebral organoids and the trophozoites were differentially labelled using the protein-binding CMFDA (green) and lipid-binding DiD (magenta) fluorescent markers, respectively. The 60-day cerebral organoid was incubated with the CMFDA for 1 hour followed by the co-culture with the DiD-labelled trophozoites. (B) Microscopy images of cerebral organoids cultured with trophozoites. On the surface, the trophozoites had multiple cytoplasm protrusions with active movement (arrowheads, left panel). Two days later, there were many trophozoites located at a small focal break in the cerebral organoids (arrowheads, right panel). Some trophozoites remained elongated in shape (asterisks, right panel), while most were oval (arrowheads, right panel). Scale bars, 100 μm.
(TIF)

**S3 Fig. The comparison of size between normal cerebral organoids and cocultured with trophozoites.** (A) The bar chart showing the diameter of human cerebral organoids and a X-day coculture of *B. mandrillaris* trophozoites with human cerebral organoids. Each dot is represented to the cerebral organoids (n = 8). The data are the mean ± SD. The statistical analysis was performed by an independent sample t test. (B) Representative images show the cerebral organoids in control and coculture experiments (scale bars, 200 μm). In the coculture with trophozoites, the outermost layer of the cerebral organoids is not smooth (arrowheads).
(TIF)

**S4 Fig. Quantitative measurement of the amoeba and host cells on the H&E-stained sections.** Three representative microscopic fields (magnification of 100X, objective lens) of coculture organoids with *B. mandrillaris* after DMSO and nitroxoline exposure were subjected to cell counting. The number of *B. mandrillaris* is enumerated based on polymorphic and round shape (yellow circle), while that of other cells is counted based on the nucleus stain (green circle). Three independent examiners and one pathological expert counted the number without

knowing the sample types to avoid bias (n = 4).
(TIF)

**S5 Fig. Cytopathology effect of cerebral organoids cocultured with trophozoites with and without miltefosine.** (A) Representative H&E staining images of cerebral organoids cocultured with trophozoites with and without 100 μM miltefosine (scale bars, 20 μm). In the DMSO-treated control (left panel), the trophozoites invaded the inner layer (arrowhead), while the meninge-like layer detached from the outermost layer (arrow). In the miltefosine-treated samples (right panel), the outermost layer of the cerebral organoids was attached by trophozoites (arrow). Some trophozoites had a large unstained space in the cytoplasm (arrowheads). (B) Representative H&E staining images of cerebral organoids in coculture with trophozoites and 0.4% DMSO. (C) The hollow inside the cerebral organoids after miltefosine exposure. There is no appearance of trophozoites in the hollow position.
(TIF)

**S1 Video. Z-stack image of the human cerebral organoid (green) cocultured with *B. mandrillaris* trophozoites (magenta). Representative fluorescence image is displayed in X-Y-Z axis.**
(MP4)

**S2 Video. Fluorescence confocal image of the human cerebral organoid (green) cocultured with *B. mandrillaris* trophozoites (magenta). Representative image is displayed from top to bottom area of the organoid.**
(MP4)

**S3 Video. Live imaging image of the human cerebral organoid cocultured with *B. mandrillaris* trophozoites under an inverted light microscope.**
(MP4)

**S1 Table. Data statistical analysis of up- and down regulation genes.**
(XLSX)

**S2 Table. List of top 100 highly expressed genes.**
(XLSX)

## Acknowledgments

We would like to thank Dr. Methichit Wattanapanitch for providing the human iPSC line MUSli001-A, Dr. Christopher A. Rice for providing nitroxoline and miltefosine, and Nitirat Panadsako for the initial data analysis. Support for fluorescence microscopy analysis was provided by staff from Olympus Thailand. Technical assistance was provided by all members at the Siriraj Center of Regenerative Medicine and Siriraj Integrative Center for Neglected Parasitic Diseases.

## Author Contributions

**Conceptualization:** Porntida Kobpornchai, Kasem Kulkeaw.

**Data curation:** Nongnat Tongkrajang, Porntida Kobpornchai, Pratima Dubey, Urai Chaisri.

**Formal analysis:** Nongnat Tongkrajang, Porntida Kobpornchai, Urai Chaisri.

**Funding acquisition:** Kasem Kulkeaw.

**Methodology:** Nongnat Tongkrajang, Porntida Kobpornchai, Pratima Dubey.

**Supervision:** Kasem Kulkeaw.

**Visualization:** Nongnat Tongkrajang, Porntida Kobpornchai, Pratima Dubey, Urai Chaisri.

**Writing – original draft:** Nongnat Tongkrajang, Porntida Kobpornchai, Kasem Kulkeaw.

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
