## [Decision Letter · Decision Letter 0]

7 May 2024

Dear Dr Kulkeaw,

Thank you very much for submitting your manuscript "Modelling amoebic brain infection caused by Balamuthia mandrillaris using a human cerebral organoid" for consideration at PLOS Neglected Tropical Diseases. As with all papers reviewed by the journal, your manuscript was reviewed by members of the editorial board and by several independent reviewers. In light of the reviews (below this email), we would like to invite the resubmission of a significantly-revised version that takes into account the reviewers' comments. 

We cannot make any decision about publication until we have seen the revised manuscript and your response to the reviewers' comments. Your revised manuscript is also likely to be sent to reviewers for further evaluation.

Sincerely,

Jaap J van Hellemond

Academic Editor

Claudia Brodskyn

Section Editor

Reviewer's Responses to Questions

**Key Review Criteria Required for Acceptance?**

**Methods**

-Are the objectives of the study clearly articulated with a clear testable hypothesis stated?

-Is the study design appropriate to address the stated objectives?

-Is the population clearly described and appropriate for the hypothesis being tested?

-Is the sample size sufficient to ensure adequate power to address the hypothesis being tested?

-Were correct statistical analysis used to support conclusions?

-Are there concerns about ethical or regulatory requirements being met?

Reviewer #1: (No Response)

Reviewer #2: Study design and methods are generally appropriate and well described. 

It is not clear why expression level for some of the genes of interest required quantitative PCR, while Illumina NGS and bioinformatics analysis was suitable for others, as described in lines 198-204.

Reviewer #3: The purpose of the study is clearly stated with a clear and testable hypothesis.

-The study design is appropriate to meet the stated objectives.

-Accurate statistical analysis was used to support the results.

-There are no concerns about meeting ethical or regulatory requirements.

**Results**

-Does the analysis presented match the analysis plan?

-Are the results clearly and completely presented?

-Are the figures (Tables, Images) of sufficient quality for clarity?

Reviewer #1: (No Response)

Reviewer #2: In the Figure 1 legend, line 298 refers to 'regions of interest (ROI) are outlined in the left panel'. This panel seems to have been omitted.

In Figure 2, apparent degeneration of amoebae in the hollow interior of the treated cerebral organoids was shown, but not the corresponding image of untreated control organoids; so it is not clearly demonstrated that changes were due to treatment, as vacuolation could be a tissue-processing artefact. Also, what were the morphological changes induced by miltefosine - were they similar to those shown in 2D & 2E?

Miltefosine is recommended for treatment of B. mandrillaris infection; it would be helpful to show the dose-response curve for miltefosine alongside or together with that of nitroxoline - it could be on the same graph since a log scale is used. 

In the Fig. 2 legend, the meaning of arrows and arrowheads for 2D should also be specified here.

Figure 3: 3B heatmap shows 6 columns of transcripts, but it is not specified which are the 2 controls and 2 tests (line 404 in the legend). For the non-expert reader, 3C is difficult to understand, likewise 3E - what is the difference between the left and right sides of the image?

Although mentioned in the text, legend for Fig 3 should also specify what 3D shows.

Reviewer #3: The analysis presented matches the analysis plan.

-The results are presented clearly and completely.

-Figures (Tables, Visuals) are not of sufficient quality in terms of understandability. Readability and quality should be increased.

**Conclusions**

-Are the conclusions supported by the data presented?

-Are the limitations of analysis clearly described?

-Do the authors discuss how these data can be helpful to advance our understanding of the topic under study?

-Is public health relevance addressed?

Reviewer #1: (No Response)

Reviewer #2: Discussion and conclusions are otherwise appropriate. Regarding line 452, in the present study it would have been interesting to investigate whether brain damage markers were reduced in nitroxoline-treated organoids, compared with untreated ones. Line 479 has relevance to this question.

Reviewer #3: -The results are supported by the data presented.

-Limitations of the analysis are not clearly defined.

**Editorial and Data Presentation Modifications?**

Reviewer #1: (No Response)

Reviewer #2: See suggestions in the Results review.

Line 35 or 111: show abbreviation iPSCs, as this is the first use.

Line 76: change to ...antiparasitics but are largely ineffective.

Line 146: show provenance of the drugs.

Line 201-202: repetition of information.

Line 248: delete 'that of' 

Line 287: The data are the mean of triplicates of the relative expression level +-SD. Delete next sentence.

Line 344: Insert 'treated' in front of 'cerebral'.

Line 355: Replace 'lining' with 'covering'.

Line 358: Specify meaning of arrows in 2A(c).

Line 394: Remove comma after 'pathways'.

Line 424: Provide reference no.

Lines 557, 573, 654, 655: italicise scientific names.

Line 633: change to 'specimens'.

Reviewer #3: (No Response)

**Summary and General Comments**

Reviewer #1: This is an important paper that will help further research on a very neglected serious pathogen. The authors have succeeded in developing a useful and sophisticated model for studying host-amoeba interactions. However, I have one major and several minor comments that I think need to be considered.

My main criticism concerns the effect of nitroxoline on amoeba-induced organoid damage. First, it is not clearly described in the results, and second, there is only one image each with and without the drug, which is uninformative. More images are needed, ideally some quantification. It is also unclear why the samples were analyzed 4 days after drug exposure, rather than 16 days. After a longer period of time, would the infection have developed regardless of the drug effect? What would happen if the drug was administered to an already established infection rather than at the beginning of the coculture, which is what the clinical cases are all about?

Minor:

In both the abstract and the author summary, the authors claim that the use of human cerebral organoids (helps) unravel the host response. I would not say this, because while this model is certainly important for understanding the mechanisms of amoeba pathogenicity, it lacks, as the authors themselves admit, the complexity of the host, especially the immune system.

Lines 93-95. Text flow is a bit confusing.

Line 80 „host‒pathogen interaction“. Which pathogens?

Line 76. should be „antiparasitics“

Line 141 „when cysts appeared at less than 1%“ could you please explain?

Line 160 CO2. subscript

Lines 210-211. is there a reference for these markers?

Line 248 „FOXG1 is a transcription factor that of essential for forebrain development“ sentence structure error?

Line 272. can you explain how Fluo-4 works?

Lines 329-330 „None of the chemokine (IL-8), inflammatory regulator (IL-1b and TNF-a),

or immunostimulatory IFN-g transcript levels increased in the coculture.“ In fact, they have significantly decreased. Could you please explain why and what the consequences would be?

Lines 344-347. please write which conditions (treated/untreated).

Fig. 2E. why only showing treated sample?

Lines 461-463. please write how CCL7 is changed in your data

Line 478 „Further validation with clinical samples from BAE patients has highlighted the utility of these biomarkers for early diagnosis and for studying response to treatment.“ Is it true? Do you have a reference for this statement?

S2 Fig. Please provide better description in legend: which time points, CMDFA/DiD staining explanation. You don't mention the middle panel and I don't see asterisks in the right panel. 

Transcriptomics: Shouldn't there be some more detailed data in the supplement?

Reviewer #2: Interesting, novel and generally well-written. GAE is an emerging disease and although rare, is usually fatal. Any promising advance in infection modelling and treatment is important.

Reviewer #3: First of all, I want to thank to all of the authors of the manuscript. Granulomatous amebic encephalitis, a central nervous system infection caused by Balamuthia mandrillaris, is a life-threatening infection. There are very few cases diagnosed and treated with this infection. There is a significant gap in both the treatment and diagnosis of this disease. Filling these gaps with the creation of such new models will offer new treatment and diagnostic options. 

This study is a good and comprehensive study. However, it would be better to increase the resolution of the figures to make them readable.

PLOS authors have the option to publish the peer review history of their article (what does this mean?). If published, this will include your full peer review and any attached files.

Reviewer #1: No

Reviewer #2: Yes: John A. Frean

Reviewer #3: No
---

## [Decision Letter · Decision Letter 1]

6 Jun 2024

Dear Dr Kulkeaw,

We are pleased to inform you that your manuscript 'Modelling amoebic brain infection caused by Balamuthia mandrillaris using a human cerebral organoid' has been provisionally accepted for publication in PLOS Neglected Tropical Diseases.

Best regards,

Jaap J van Hellemond

Academic Editor

Claudia Brodskyn

Section Editor

Reviewer's Responses to Questions

**Key Review Criteria Required for Acceptance?**

**Methods**

-Are the objectives of the study clearly articulated with a clear testable hypothesis stated?

-Is the study design appropriate to address the stated objectives?

-Is the population clearly described and appropriate for the hypothesis being tested?

-Is the sample size sufficient to ensure adequate power to address the hypothesis being tested?

-Were correct statistical analysis used to support conclusions?

-Are there concerns about ethical or regulatory requirements being met?

Reviewer #1: (No Response)

Reviewer #2: As in Round 1 comments

Reviewer #3: (No Response)

**Results**

-Does the analysis presented match the analysis plan?

-Are the results clearly and completely presented?

-Are the figures (Tables, Images) of sufficient quality for clarity?

Reviewer #1: (No Response)

Reviewer #2: My comments on this aspect have been adequately addressed

Reviewer #3: (No Response)

**Conclusions**

-Are the conclusions supported by the data presented?

-Are the limitations of analysis clearly described?

-Do the authors discuss how these data can be helpful to advance our understanding of the topic under study?

-Is public health relevance addressed?

Reviewer #1: (No Response)

Reviewer #2: My comments on this aspect have been adequately addressed

Reviewer #3: (No Response)

**Editorial and Data Presentation Modifications?**

Reviewer #1: (No Response)

Reviewer #2: Line 91: The use of English in this sentence requires improvement. The reference to the previous sentence is clear, so I suggest replacing the first phrase with 'In this in-vitro model, Zika....'etc. Alternatively, omit the first phrase altogether.

Reviewer #3: Figures have been corrected and the quality has been improved.

**Summary and General Comments**

Reviewer #1: I am satisfied with the revised version of the paper and wish the authors good luck in their future research.

Reviewer #2: The revised manuscript is much improved. My concerns were satisfactorily addressed.

Reviewer #3: Since the previous first version of this study was reviewed, the requested corrections and changes were made by the authors.

PLOS authors have the option to publish the peer review history of their article (what does this mean?). If published, this will include your full peer review and any attached files.

Reviewer #1: **Yes: **Robert Sutak

Reviewer #2: **Yes: **John A. Frean

Reviewer #3: No

---

## [Editor Report · Acceptance letter]

17 Jun 2024

Dear Dr Kulkeaw,

We are delighted to inform you that your manuscript, "Modelling amoebic brain infection caused by *Balamuthia mandrillaris* using a human cerebral organoid," has been formally accepted for publication in PLOS Neglected Tropical Diseases.

Best regards,

Shaden Kamhawi

co-Editor-in-Chief

Paul Brindley

co-Editor-in-Chief
